

# CAPN: a Combine Attention Partial Network for glove detection

Feng Yu[1,2], Jialong Zhu[1], Yukun Chen[1], Shuqing Liu[1] and Minghua Jiang[1,2]

[1] School of Computer Science and Artificial Intelligence, Wuhan Textile University, Wuhan, Jiangxia District, China
[2] Engineering Research Center of Hubei Province for Clothing Information, Wuhan, Jiangxia District, China

## ABSTRACT

Accidents caused by operators failing to wear safety gloves are a frequent problem at electric power operation sites, and the inefficiency of manual supervision and the lack of effective supervision methods result in frequent electricity safety accidents. To address the issue of low accuracy in glove detection with small-scale glove datasets. This article proposes a real-time glove detection algorithm using video surveillance to address these issues. The approach employs transfer learning and an attention mechanism to enhance detection average precision. The key ideas of our algorithm are as follows: (1) introducing the Combine Attention Partial Network (CAPN) based on convolutional neural networks, which can accurately recognize whether gloves are being worn, (2) combining channel attention and spatial attention modules to improve CAPN's ability to extract deeper feature information and recognition accuracy, and (3) using transfer learning to transfer human hand features in different states to gloves to enhance the small sample dataset of gloves. Experimental results show that the proposed network structure achieves high performance in terms of detection average precision. The average precision of glove detection reached 96.59%, demonstrating the efficacy of CAPN.

## INTRODUCTION

With the steady increase in demand for electricity due to global modernization, there has been a proliferation of power infrastructure. It is necessary for electrical equipment to be regularly maintained and repaired, and staff must wear gloves correctly to safely conduct electrical operations. However, manual supervision is insufficient as it cannot supervise glove usage in real-time, leading to possible negligence and failing to achieve the desired safety outcome. With the advancement of computer vision (*Esteva et al., 2021*), object detection technology using video surveillance has been implemented in various environments (*Liu et al., 2022*), such as power plants and electrical substations. This technology is highly robust and can accurately identify operators wearing gloves.

Object detection can be broadly classified into two types: traditional image processing methods (*Sarker, 2021*; *Guo, Zhang & Tang, 2021*) and deep neural network methods. The

Corresponding author
Minghua Jiang, minghuajiang@wtu.edu.cn

traditional approach involves four stages: image preprocessing, target area selection, feature extraction, and classifier selection. In the preprocessing stage, the objective is to remove irrelevant information from the image. During target area selection, sliding windows of varying sizes are used to scan the entire image and extract texture and shape features. These features are then represented as vectors, which are passed through the feature classifier to obtain the probability of classification. In the case of deep convolutional neural network methods, the target detection problem is transformed into a regression problem for locating the target frame and calculating the category probability. The entire image is divided into a fixed number of grid cells, and each cell predicts whether an object is present at that location and determines the coordinates and size of the object's bounding box. Convolutional neural networks are used for feature extraction, which can extract more robust feature information when compared to traditional algorithms. A lightweight multi-scale network (LMSN) (*Li, Li & Zhou, 2022*) is proposed to utilize multi-scale information to enhance semantic information interaction at each scale. Secondly, through a lightweight receiving domain enhancement module, the feature extraction capability of the network is enhanced, which helps to better identify small targets. The YOLOV4 architecture is used to construct an effective hand detection method, and built Cross Stage Partial (CSP) and Spatial Pyramid Pooling (SPP) layer network modules (*Dewi & Juli Christanto, 2022*), which are more capable of better detection of hands in motion and improved detection average precision. An accurate, flexible, and completely anchor-free target detection framework (*Kong et al., 2020*) is proposed, which can generate category-independent bounding boxes for locations that may contain object information, which can make the model more accurate. The above-mentioned detection algorithms can identify the basic characteristics of gloves, but in the actual electric power work site, it is necessary to accurately identify whether the operator is wearing safety gloves. There are also some difficulties: (1) the object of the operator wearing gloves is small, and it is difficult to extract the characteristics of the gloves in depth, (2) traditional object detection algorithms are affected by different environments (such as light, raindrops, smog), result in false detection and missing detection, and (3) the data set of gloves is lacking and difficult to obtain.

To address the aforementioned issues in glove detection in electric power scenario, this article proposes a convolutional neural network (CNN) algorithm based on transfer learning and attention mechanism to improve precision. The network comprises three parts: the backbone network, the feature pyramid network, and the classification regression prediction component. In the backbone network, residual modules and convolutional modules are stacked to perform convolutional operations on input data and convolutional kernels to extract feature information, which can effectively reduce the number of network parameters and computation. The residual module adds the input feature map directly to the feature map after the non-linear transformation to obtain the residual feature map. By introducing skip connections, the residual module can effectively alleviate the problem of vanishing and exploding gradients, thus enhancing the network's feature extraction capability for gloves. The feature pyramid network introduces different levels of feature information from the backbone network and uses channel spatial attention mechanism (*Chaudhari et al., 2021*) to adjust the weight of feature maps by focusing on

different channels and spatial information, making the network more attentive to the glove features. In the prediction component, classification and bounding box regression techniques are used to identify the target. During the overall network's training process, transfer learning (*Zhuang et al., 2020*) is utilized to transfer the characteristics of human hands to glove recognition. Experimental results demonstrate that the proposed method outperforms existing approaches in detecting gloves in electric power scenario with improved average precision and efficiency.

The remainder of this article is structured as follows: the Related work section discusses the related work to our proposed method. The Glove detection method section presents the proposed method and its underlying theory. The Experiments section presents the experimental results and analysis. The final section concludes the article and outlines future work.

## RELATED WORK

Related work mainly includes three parts: (1) object detection methods based on deep learning, (2) transfer learning, and (3) attention mechanism.

### Object detection method based on deep learning

Object detection is one of the most classic problems in computer vision (*Li et al., 2021*; *Kattenborn et al., 2021*). In object detection networks (*Cengil, Çinar & Yildirim, 2021*), features are extracted using convolutional neural networks, which have been widely applied in fields such as classification (*Wang & Chen, 2019*; *Shahverdy et al., 2020*), object detection, and object segmentation. Currently, there are different methods for glove detection, with most methods using sensors to detect whether gloves are being worn (*Barfidokht et al., 2019*). In some dangerous situations, object detection is rarely used to identify workers wearing gloves. Object detection networks can identify workers wearing gloves to improve safety in dangerous areas. Currently, object detection networks can be divided into two types: two-stage object detection algorithms and one-stage object detection algorithms.

#### Two-stage object detection

Two-stage object detection algorithms typically consist of two stages. The first stage is to generate region proposals, which involves identifying candidate regions in the input image that may contain objects. The second stage involves classification and position regression of the region proposals to obtain the final object detection results. A variant of the RCNN algorithm is proposed for two-stage object detection (*Cheng et al., 2018*), which is widely applied in traditional computer vision. Firstly, 2,000 region proposals are randomly generated in the input image. Secondly, these candidate frames are sent to the support vector machines (SVM) (*Cherkassky & Ma, 2004*) classifier for object detection. Finally, the candidate frames are regressed to refine the object localization, and the output is the result of target detection. Fast R-CNN (*Girshick, 2015*) and Faster R-CNN are both improvements on the R-CNN method, achieving better results for object detection due to their ability to share convolutional features and introduce region proposal networks.

There are also two other common two-stage object detection networks, namely Mask R-CNN (*He et al., 2017*) and R-FCN (*Dai et al., 2016*). Mask R-CNN adds instance segmentation to Faster R-CNN, allowing it to simultaneously detect objects and generate accurate segmentation masks. R-FCN is a more efficient two-stage object detection network that reduces computation by sharing convolutional features across the global context and achieves position invariance through region pooling.

### *One-stage object detection*

One-stage object detection algorithms detect objects by directly predicting their class and position on the feature maps, without generating region proposals, which leads to faster detection speeds compared to two-stage object detection. Therefore, one-stage object detection algorithms enable end-to-end object detection. the YOLO v1-v5 (*Redmon et al., 2016*; *Redmon & Farhadi, 2017*; *Tian et al., 2019a*; *Bochkovskiy, Wang & Liao, 2020*; *Zhu et al., 2021*) series is considered the most representative. These algorithms abandon traditional feature extraction after object positioning and instead extract features directly for classification, regression, and prediction (*Tian et al., 2019b*). They use fully connected layers in classification and regression to detect objects, which greatly speeds up the detection process. Another method (*Ismail et al., 2021*) involves using images obtained from real-time videos through the cascade classifier Haar to detect the target hand through the Region Of Interest (ROI) theory. This method can be applied to detect objects in the human body. To construct an effective hand detection method, the YOLOV4 architecture (*Dewi & Juli Christanto, 2022*) is used and CSP and SPP layer network modules are built, making it more capable of detecting hands in motion and improving detection precision. Additionally, the YOLOX's anchor-free architecture (*Ferdous & Ahsan, 2022*) is used to detect personal protective equipment in the construction industry, which greatly improves the safety of operators.

### Transfer learning

Transfer learning is a subfield of machine learning (*Hart et al., 2021*) that involves retaining useful information from previously learned data to solve new problems faster and more effectively. Due to the lack of data in specific tasks, transfer learning is widely used in image classification and object detection experiments. Many studies have shown that using pre-trained convolutional neural networks (CNNs) for transfer learning produces better results than training from scratch. Therefore, an object detection framework for deep CNNs is important in the base network. For instance, GoogleLeNet (*Szegedy et al., 2015*), VGG Network (*Simonyan & Zisserman, 2014*), and ResNet (*He et al., 2016*) all utilize pre-training weights from the ImageNet dataset and demonstrate superior performance. To detect micro-aperture radar, an object detection method utilizing transfer learning is proposed, which extracts features from the three-channel sub-aperture (*Wang et al., 2018*) data obtained from the existing Synthetic Aperture Radar (SAR) target recognition dataset and the Moving and Stationary Target Acquisition and Recognition (MSTAR) dataset. The Singer Shot Detector (SSD) (*Liu et al., 2016*) algorithm is then used to detect the target, resulting in improved detection performance. This approach enables end-to-end trainable

aircraft detection with a single deep CNN and limited training samples (*Chen, Zhang & Ouyang, 2018*), demonstrating high average precision and significant application potential for remote sensing target detection.

### Attention mechanism

Attention mechanisms are often used in neural network structures to better extract key information while ignoring unimportant information. There are three common attention mechanisms: spatial attention, channel attention, and self-attention mechanisms. Point-Wise Spatial Network (PSANet) (*Zhao et al., 2018*) is proposed to reduce local neighborhood constraints. Each location on the feature map is connected to all other locations through an adaptive learned attention mask, and this approach achieved good results on the ADE20K (*Zhou et al., 2017*) and PASCAL VOC 2012 datasets. Semantic Segmentation Network with Spatial and Channel Attention (SCAttNet) (*Li et al., 2020*) is proposed for high-resolution remote sensing image segmentation, and it has achieved significant improvement. Squeeze-and-Excitation Networks (SE-Net) (*Hu, Shen & Sun, 2018*) are proposed to adaptively calibrate the feature responses of channel directions, forming a SENet architecture that efficiently generalizes across different datasets and further improves the feature extraction ability of CNNs. The self-attention module (*Vaswani et al., 2017*) is initially used in the field of natural language processing (*Liu et al., 2021*). It uses a fixed-length vector in the image field, and then applies keys, queries, and values to perform global modeling, thus improving the global information extraction ability. The Convolutional Block Attention Module (CBAM) (*Woo et al., 2018*) is proposed, which sequentially performs the attention mechanism along two independent dimensions of channel and space. Then, the attention map is multiplied by the input feature map for adaptive feature refinement. This article also makes reasonable use of this module to improve the feature extraction of gloves, which in turn improves recognition precision.

## GLOVE DETECTION NETWORK

The main task of the algorithm proposed in this article is to quickly and accurately identify whether workers are wearing gloves at electrical worksites. The overall flow chart of the algorithm is shown in Fig. 1. This article is based on the YOLO (You Only Look Once) object detection network. In the training phase, improvements are made to the feature enhancement network, which led to the expected results being achieved.

### CAPN Framework

The feature extraction network module in the CAPN structure can effectively extract gloves during glove detection, and the improved CSP-Darknet53 (*Wang et al., 2020*) is used as the backbone network of CAPN to extract features from the input image. As shown in Fig. 1, the CBL and CSP modules are used in the feature extraction module. The CBL module is composed of ordinary convolution and Deep-wise (DW) (*Howard et al., 2019*) convolution. The DW convolution module is first applied in the MobileNet network structure, where the DW convolution performs convolution operations on each input channel separately and uses a $1 \times 1$ convolution on the output channel to integrate the

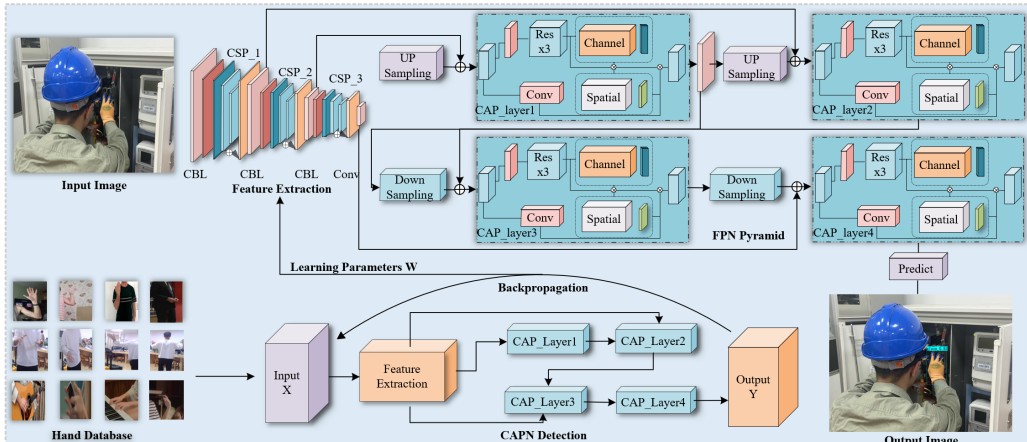

**Figure 1** **An overview of our proposed method.** The overall process is divided into two parts. The first part uses CAPN to pre-train the hand dataset to obtain the learning weight W, the second part is to apply the weight W to the detection of gloves.

results of all input channels. The CSP module, inspired by the residual module, constructs a large residual block to deepen the network, prevent gradient vanishing, and extract deeper feature information. The combination of CBL and CSP modules achieves higher precision and detection speed. Therefore, the efficient backbone network model speeds up calculation speed and reduces training parameters, making it possible to detect workers wearing gloves in real-time at the electric power operation site.

In the feature extraction network structure of CAPN, three CBL modules are used, in which the CBL module is a network structure composed of DW convolution and ordinary convolution. After each CBL module, a large residual CSP module is used. The overall design idea is to use DW convolution to independently perform a $1 \times 1$ convolution operation on the information channel of each input layer, the feature map of each feature channel will be obtained by using the ordinary convolution module to perform feature extraction again, and finally all the feature maps will be added. Owing to the DW convolution is to perform feature extraction on the information on each individual channel, the information across channels (*Rezaee et al., 2021*) is not considered, so the advantages of ordinary convolution are used to complement each other, so as to better extract the features of the image. This article use the CSP module that can deepen the network depth to extract information from the shallow feature network. This module first passes through a common convolution module, and then stacks residual modules. This residual module is composed of a convolution module, CBL module and a residual block. Finally, the output feature information is added to the CSP input feature information to form a large residual block. This module can extract the deep information of the image, which is divided into two parts. One part enters the residual block to improve the operation speed, and the other part is added to the feature map after feature extraction to fuse the information, while extracting deeper feature information, the information of the original feature map is preserved and fused, which effectively improves the extraction of image information.

Convolution operations are used in the entire feature extraction network frequently, and Batch Normalization (*Ioffe & Szegedy, 2015*) is used after each convolution to make the overall data satisfy the distribution law with a mean value of 0 and a variance of 1, and use the Sigmoid Linear Unit (SiLu) activation function for unified processing.

The feature pyramid layer (*Ghiasi, Lin & Le, 2019*) in the CAPN fuses information from different levels in the backbone network. First, the output layer in the feature extraction network is up-sampled, and the resulting feature map after passing through the CAP module is combined with the second CBL feature layer in the backbone network. This feature map is then convoluted again and subjected to feature fusion with the first CBL feature layer in the backbone network after up-sampling and CAP module features. Second, after the obtained feature map passes through the CAP module, a downsampling operation is performed immediately, and it is fused with the convolutional features in the FPN network. Finally, after downsampling and the first CBL feature layer in the backbone network, the feature fusion is sent to the CAP module to obtain the prediction result. The CAP module is a model component used for image processing, consisting of the CSP module, channel attention module, and spatial attention module. The CSP module is a convolutional neural network module used to extract features from images. It divides the input feature map into two parts, processes them differently, and then merges them. This separation and merging method can effectively reduce the number of parameters. Both the channel attention module and the spatial attention module are attention mechanisms. The channel attention module is used to adaptively adjust the weights of different channels in the input feature map to better capture the important features of input data. The spatial attention module is used to adaptively adjust the weights of different positions in the input feature map to better capture the spatial information of input data. Through the combination of these modules, the CAP module can better extract deeper information from image data. It can improve the training efficiency and generalization performance of the model, leading to better results in image processing tasks. By using the FPN pyramid, the feature layers from different levels in the backbone network can be fused, effectively combining the deep and shallow information and retaining image information on the original feature layer, which improves the recognition precision of the network. The CAPN structure is better explained in Table 1.

## CAP Attention mechanism

To address the challenge of diverse glove types and small glove sizes during glove detection at electric power work sites, a solution is proposed to extract deeper information from the input image for accurate learning of small targets such as gloves. CAP attention module is proposed to be added between convolutional layers to enhance the network's ability to extract image features, as shown in Fig. 2. The module undergoes a convolution operation, followed by three small residual modules, and finally an attention mechanism module, which combines channel attention and spatial attention to selectively amplify informative features and suppress less informative ones.

The channel attention module has two branches: average pooling and global pooling (*Chen et al., 2021*). Each branch generates a (C, 1, 1) weight vector. Average pooling,

**Table 1 The structure of the CAPN.**

| Stage | Operator | Input shape | Channel | Layer |
|---|---|---|---|---|
| 1 | Input | 640 ×640 | 3 | Input |
| 2 | CBL-1 | 640 ×640 | 3 | Feature Extraction |
| 3 | CSP-1 | 320 ×320 | 64 | Feature Extraction |
| 4 | CBL-2 | 320 ×320 | 64 | Feature Extraction |
| 5 | CSP-2 | 160 ×160 | 128 | Feature Extraction |
| 6 | CBL-3 | 160 ×160 | 128 | Feature Extraction |
| 7 | CSP-3 | 80 ×80 | 128 | Feature Extraction |
| 8 | Conv | 20 ×20 | 256 | Feature Extraction |
| 9 | Up Sampling | 40 ×40 | 256 | FPN pyramid |
| 10 | CAP-layer1 | 40 ×40 | 256 | FPN pyramid |
| 11 | Convolution | 40 ×40 | 128 | FPN pyramid |
| 12 | Up Sampling | 80 ×80 | 128 | FPN pyramid |
| 13 | CAP-layer2 | 80 ×80 | 256 | FPN pyramid |
| 14 | Down Sampling | 40 ×40 | 128 | FPN pyramid |
| 15 | CAP-layer3 | 40 ×40 | 256 | FPN pyramid |
| 16 | Down Sampling | 20 ×20 | 256 | FPN pyramid |
| 17 | CAP-layer4 | 20 ×20 | 512 | FPN pyramid |

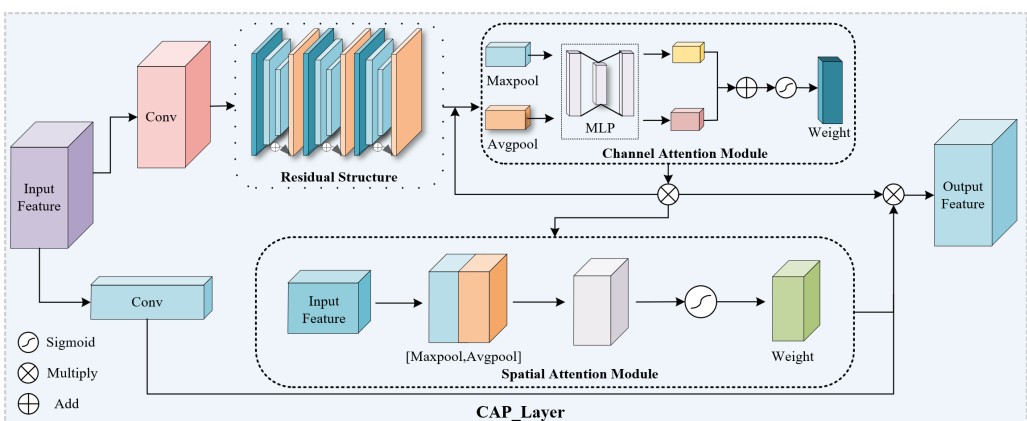

**Figure 2 The CAP attention module proposed in this article consists of channel attention module and spatial attention module.**

which slides a window over the feature map, preserves background information and reduces overfitting (*Ying, 2019*). Maximum pooling extracts feature textures and reduces the influence of irrelevant information. The weight vectors from both branches are fully connected and processed by the ReLU activation function before being added. Finally, the resulting vector is mapped to the dimensional information of each channel. This process preserves the image background information and extracts texture features simultaneously. The calculation formula for this process is shown in Eq. (1):

$$\text{Chanel}_c = \sigma(MLP(AvgPool(F)) + MLP(MaxPool(F))) = \sigma(W_1(W_0(F_{avg}^c))$$

$$+W_1(W_0(F(c_{\text{Max}}))))\tag{1}$$

where $\sigma$ is the Sigmoid activation function, MLP is the weighted layer of the fully connected layer, $W_0$ and $W_1$ correspond to the weight parameters of the fully connected layer respectively.

After multiplying the output result with the input feature vector, it is passed through a spatial attention module. The feature maps then undergo average pooling and maximum pooling, respectively, to obtain two weight vectors of size (1, H, W). The number of channels changes from C to 1, and the two weight vectors are stacked to form weight information of size (2, H, W) after convolution processing. This weight information is then multiplied with the original feature map, where each point on the original feature map is given a weight change. This process amplifies the values with larger weights in the original feature map and emphasizes the importance of information, as shown in Eq. (2):

$$\text{Spatial}_c = \sigma(f^{7\times7}(\text{concat}(AvgPool(F), MaxPool(F)))) = \sigma(f^{7\times7}([F^s_{avg}; F^s_{max}]))\tag{2}$$

where $\sigma$ is the Sigmoid activation function, $f^{7\times7}$ means that the size of the convolution kernel used in the convolution process is $7\times7$, concat is the channel splicing of the weight vector obtained after global pooling and average pooling.

Finally, after being activated by the Sigmoid function, a multiplication operation is performed with the original input feature to form a large residual edge output. The CAP module helps to represent important information in feature maps and improve detection precision.

## Transfer learning

The CAPN model is employed in this article to detect glove objects, requiring a large number of training samples. However, the availability of such training samples is limited for the specific glove detection task, leading to the challenge of learning a good CAPN model. One reason for the lack of sufficient training samples is the high cost and difficulty of collecting and annotating glove images in real-world electric power scenarios.

In this article, a data augmentation method suitable for object detection in gloves is proposed. To address this issue, several data augmentation techniques, such as random cropping, rotation, and flipping, are explored to generate additional training samples and increase the diversity of the dataset. Transfer learning from other related tasks or domains, such as hand detection or object detection in general, is also considered to pretrain the CAPN and improve its initialization and feature representation. Alternatively, synthetic data generated by computer graphics or simulation tools may be used to augment the training set and increase the variability of the images.

Transfer learning is utilized in this study to extract useful knowledge from a source dataset and apply it to the target task. The original model is trained on the dataset of human hand features using CAPN. The learned model parameters $W$ are then used as the starting point for training on glove data, without performing reverse gradient updates in the backbone network. The goal is to transfer knowledge from the source domain of human hands to the target domain of gloves.

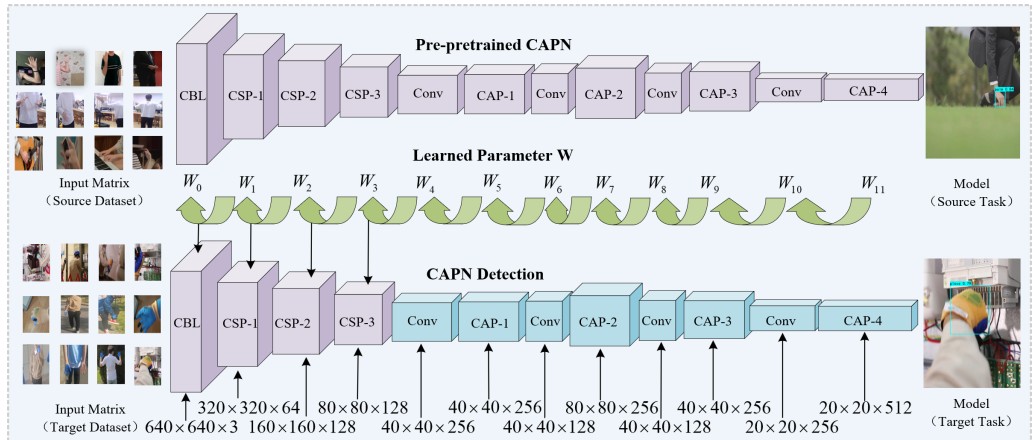

**Figure 3** The weight W learned by continuous backpropagation using the source domain dataset is used in the backbone network of the CAPN.

To achieve this, the feature information of human hands is used to learn the feature information of gloves. In the network, X and Y are used as the input and output of the original network. Let $S^m(x)$ be the feature expression of the mth layer in the pre-trained source network and $T_w$ be the network to be learned in the target domain. Then $T_w^n(x)$ is the feature expression of the n-th layer in the target domain network, where $w$ is the set of parameters to be learned, as shown in the following Equation (3):

$$\| r_w(T_w^n(x)) - S^m(x) \|_2^2 \tag{3}$$

where $r_w$ is a linear transformation of the input matrix, the m-th layer and n-th layer in the above formula perform weight transfer, which is the overall transfer learning goal.

Considering that not all layer features in the source domain are conducive to the target task, so a weight matrix, $\gamma^{m,n} > 0$ is proposed to represent the transferable index of the m-th layer in the source domain to the n-th layer in the target domain, the larger metric, the more migrations can be made, and the smaller metric, the less migrations, and parameters can be expressed as Equation (4):

$$\gamma^{m,n} > 0 = g_\theta^{m,n}(S^m(x)) \tag{4}$$

where the $\theta$ parameter is the learning target. The weight of the network that can be migrated according to the index of the weight matrix. As shown in Fig. 3, this training process enables the efficient transfer of knowledge acquired from source datasets, thereby improving the learning ability of target tasks using relatively small datasets, reduces the negative impact of the small number of datasets significantly, and improved network extraction of gloves features.

## EXPERIMENTS

This article presents a series of experiments conducted on a Windows 10 platform system equipped with an NVIDIA RTX3050 graphics card. Prior to training, all images are resized

to 640×640 and subjected to data augmentation techniques, such as random horizontal flipping and resizing.The following software packages and versions are utilized: Python 3.9.12, PyTorch 1.12.1, torchvision 0.13.1, numpy 1.21.5, and OpenCV 4.5.1. The sgd optimizer is employed to optimize the model, with an initial learning rate of 0.01 and a decay coefficient of 0.0005. A batch size of 16 is used, and the model is trained for 300 iterations before reducing the learning rate to 0.0004 for an additional 100 iterations.The performance of our object detection model is evaluated against several state-of-the-art models, including Faster-RCNN, SSD, YOLOV3, YOLOV4, YOLOV5, YOLOX (*Ge et al., 2021*), and YOLO V7 (*Wang, Bochkovskiy & Liao, 2022*), using standard evaluation metrics such as average precision (AP), precision, recall, F1 score, and log-average miss rate. Our model achieved an AP of 96.59%, outperforming all other models.

## Data set

The experimental data includes two self-made datasets: the hand dataset and the gloves dataset. The hand dataset comprises 8,000 natural-state images of human hands captured in various environments. Among these, 7,200 images are used for training, 720 images are used for validation, and 800 images are used for testing the source domain data model. The hand dataset is obtained from hand captures in various scenarios, which effectively ensures the state characteristics of hands in different postures. The gloves dataset is obtained from outdoor and indoor field shooting of electric power work sites, and contains 1,000 images. Of these, 900 images are used for training, 90 images are used for validation, and 100 images are used for testing the source domain data model. As obtaining datasets in power operation scenarios is difficult and requires consideration of privacy concerns, we created a small dataset consisting of 1,000 samples. The uniqueness of these datasets contributes to their significance and theoretical value in the research.

## Pre-training

In this study, the hand recognition model is first pre-trained using a hand dataset, and the pre-trained weights are not used during the subsequent training on the hand dataset, resulting in a model capable of recognizing hands. The learned hand features are then transferred to the glove training process, as described in the transfer learning section. The platform, environment version, optimizer, and learning rate used for both training processes are consistent. Prior to network training, data augmentation (*Bayer, Kaufhold & Reuter, 2022*) operations are performed on the dataset to increase its size and diversity. Specifically, four images are randomly selected and stretching, scaling, and rotation operations are applied to a portion of each image, along with its corresponding label information. The augmented images are then recombined into a spliced image. This process enhanced the information of the detected objects and increased the batch size, thereby improving the performance of the model. The specific image enhancement operation is shown in Fig. 4. By utilizing this method, the dataset is expanded, and the background information of the detection targets is enriched, effectively improving the diversity of the dataset.

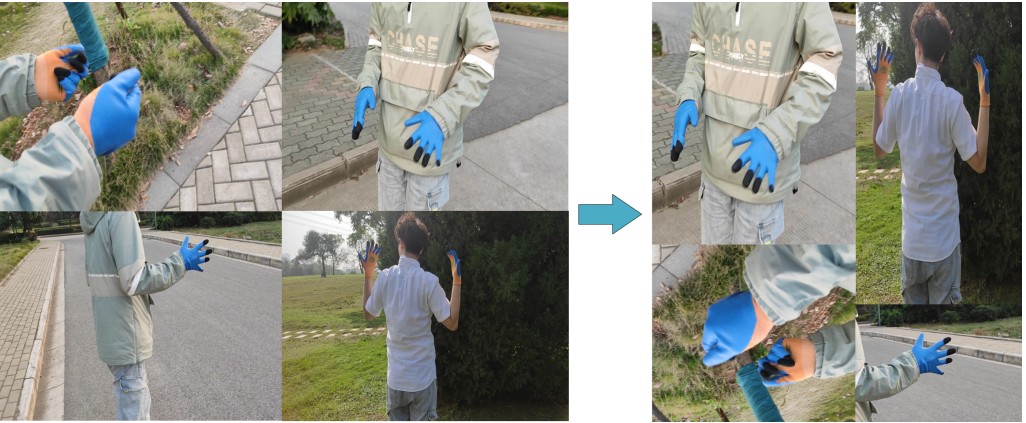

**Figure 4** **Data enhancement.** Randomly stitching four pictures together to expand the dataset.

## Evaluation criterion

The experiment uses precision rate (P), recall rate (R), F1-score (F1), average precision (AP), intersection over union (IoU), and log-average miss rate (MR-2) as the main evaluation metrics, which are defined in the following Eqs. (5), (6), (7) and (8):

$$\text{Precision} = \frac{TP}{TP + FP} \tag{5}$$

$$F1 = \frac{2 \times P \times R}{P + R} \tag{6}$$

$$\text{Recall} = \frac{TP}{TP + FN} \tag{7}$$

$$\text{MR-2}_i = \frac{1}{|\text{gt}i|}\sum j = 1^{|\text{gt}i|}\log(\max(1 - \text{IoU}(d_j, \text{gt}i, k))) \tag{8}$$

where the true positives (TP) refer to the cases where the predicted value matches the ground truth, and the predicted value is a positive sample. False positives (FP) refer to the cases where the predicted value differs from the ground truth, but the predicted value is a positive sample. False negatives (FN) refer to the cases where the predicted value differs from the ground truth, but the predicted value is a negative sample.

Precision (P) is the ratio of true positive samples to all predicted positive samples, while Recall (R) is the ratio of true positive samples to all true positive samples. The F1 score (F1) is a single metric that combines both precision and recall. Average precision (AP) is the area under the Precision-Recall curve and is a widely used performance metric for object detection. A higher AP indicates better performance. The log-average miss rate (MR-2) is a performance metric for object detection that measures the logarithmically averaged false positive rate at different levels of recall. |gt$i$| denotes the number of ground truth

annotations for the $i$th class in the dataset, $\mathrm{gt}i,k$ denotes the $k$th ground truth bounding box for the $i$th class, $d_j$ denotes the $j$th predicted bounding box, and $\mathrm{IoU}(d_j, \mathrm{gt}_{i,k})$ denotes the IoU between the $j$th predicted bounding box and the $k$th ground truth bounding box for the $i$th class.

## Comparative experiments

To evaluate the effectiveness of the CAPN algorithm for detecting gloves in power operation sites, comparison experiments are conducted with several state-of-the-art object detection algorithms, including Faster-RCNN, SSD, YOLOV3, YOLOV4, YOLOV5, YOLOX, and YOLOV7. The experiments are designed to assess the performance of each algorithm in detecting workers wearing gloves, using standard metrics such as precision, recall, F1 score ,and MR-2. Currently, YOLO X is considered one of the top-performing object detection algorithms. It utilizes an Anchor-free approach, which predicts the coordinates of each prediction center point with only four parameters, reducing the number of parameters and improving the calculation speed of the target detection algorithm. To build on this approach, an anchor-free CAPN based on the predictive head approach is proposed, which can efficiently detect workers wearing gloves in real-time in power operation sites.

The training approach for the glove dataset in this study involves first training a hand weight model using the hand dataset, and then using this weight model as a pre-training weight for training the glove dataset. The training process involves two stages. During the first 50 epochs, the weights of the backbone network are frozen to speed up the training process and prevent weight destruction. At this stage, only the feature extraction network can update its weights, and the model training involves only fine-tuning the feature extraction network to reduce GPU memory usage. In the following 250 epochs, all network weights can be updated, and the entire model participates in the training process to further improve model performance.

Table 2 shows the model experimental results of seven classic networks compared with the proposed method on the glove data set. It can be concluded that the proposed CAPN has achieved good results in precision, recall, and average precision. Detecting workers wearing gloves is important in power operation sites because it can help prevent electrical accidents and ensure the safety of workers. By using the CAPN algorithm, the precision and efficiency of glove detection can be improved, which can ultimately contribute to a safer workplace.

## Ablation experiments

To comprehensively evaluate the proposed CAPN in this article and understand the performance of each module, an ablation experiment is conducted. The goal of this experiment is to gradually split each part of the network and assess its impact on the overall performance of the model. By doing so, the contribution of each module to the network's precision and recall can be fully evaluated.

Specifically, the ablation experiment is divided into four parts. The first part excluded the transfer learning and attention mechanism from the CAPN network. In the second part, the attention mechanism is added to the network but the transfer learning mechanism

**Table 2  The result of comparative experiment.**

| Model | Resolution | Precision | Recall | F1-Score | AP(0.5) | AP(0.75) | MR-2 |
|---|---|---|---|---|---|---|---|
| Faster -RCNN | 224 ×224 | 46.80% | 92.86% | 62% | 86.69% | 18.34% | 0.29 |
| SSD | 300 ×300 | 97.22% | 55.56% | 71% | 87.51% | 27.13% | 0.26 |
| YOLO V3 | 416 ×416 | 88.66% | 68.25% | 77% | 80.19% | 13.11% | 0.39 |
| YOLO V4 | 416 ×416 | 93.07% | 74.6% | 83% | 88.57% | 18.52% | 0.26 |
| YOLO v5 | 640 ×640 | 92.56% | 88.89% | 91% | 94.28% | 40.42% | 0.13 |
| YOLO X | 640 ×640 | 90.23% | 95.24% | 93% | 94.55% | 41.97% | 0.12 |
| YOLO V7 | 640 ×640 | 93.20% | 76.19% | 84% | 90.05% | 32.56% | 0.21 |
| CAPN (ours) | 640 ×640 | 95.11% | 94.76% | 95% | 96.59% | 48.78% | 0.07 |

**Table 3  The result of ablation experiment.**

| CAPN | CAP Attention mechanism | Transfer learning | Precision | Recall | AP(0.5) |
|---|---|---|---|---|---|
| ✓ | | | 90.23% | 95.24% | 94.55% |
| ✓ | ✓ | | 92.86% | 92.86% | 94.72% |
| ✓ | | ✓ | 90.98% | 96.03% | 95.78% |
| ✓ | ✓ | ✓ | 95.11% | 94.76% | 96.59% |

is not included. In the third part, the transfer learning mechanism is added to the network but the attention mechanism is not included. Finally, in the fourth part, both mechanisms are introduced. Table 3 displays the results of the ablation experiment, which show the performance of the network with and without each of the modules.

Based on the experimental results presented in Table 3, it is observed that the introduction of the attention mechanism module improved the network's recognition precision for targets but reduced the recall of the targets. This is due to the attention mechanism's deepening of the extraction of detailed features, which resulted in the loss of some contextual information. However, it effectively improved the precision of recognizing gloves with smaller targets.

Furthermore, during the transfer learning process in this study, it is observed that the introduction of attention mechanisms did not significantly improve the recall of the target, while the recall rate is greatly enhanced. The reason for this is that transfer learning commonly utilizes pre-trained models that are trained on large benchmark datasets. As the features of hands are similar to those of gloves in the electrical work scene, the weights from the hand dataset features are effectively utilized to enhance the network's recall of gloves. This led to an improvement in recognition recall, even with a limited glove dataset of only 1,000 samples, effectively preventing missed inspections of gloves at power work sites.

After introducing both the attention mechanism and transfer learning, it is observed that there is an improvement in both precision and recall of the network. These two modules complemented each other and enhanced the precision and stability of target recognition, ensuring effective identification of gloves at the power work site. In conclusion, this article

**Table 4   The result of different seeds experiment.**

| Seed | Method | Scaling | Precision | Recall | F1-Score | MR-2 | AP(IoU:0.5) |
|------|--------|---------|-----------|--------|----------|------|-------------|
|      |        | 0.01 | 95.11% | 94.76% | 95% | 0.07 | 96.59% |
| 1 | Normal | 0.02 | 93.98% | 95.43% | 94% | 0.08 | 96.17% |
|      |        | 0.03 | 94.44% | 95.51% | 94% | 0.08 | 96.08% |
|      |        | 0.01 | 95.11% | 94.76% | 95% | 0.08 | 95.11% |
| 2 | Xavier | 0.02 | 94.10% | 95.51% | 95% | 0.08 | 95.76% |
|      |        | 0.03 | 94.78% | 95.13% | 95% | 0.08 | 96.00% |
|      |        | 0.01 | 94.42% | 95.13% | 95% | 0.09 | 95.43% |
| 3 | Kaiming | 0.02 | 92.31% | 95.11% | 95% | 0.08 | 95.85% |
|      |        | 0.03 | 93.59% | 94.86% | 95% | 0.08 | 95.46% |

utilizes the combination of attention mechanism and transfer learning to improve the average precision and stability of target recognition in the electrical work scene.

## Seed experiments

Using different random seeds to verify the stability of the network is a common practice in object detection networks. A random seed is an initial value used to generate a sequence of random numbers. In deep learning, random seeds are typically used for initializing model parameter. Therefore, different random seeds can lead to different model parameters and random operations, which can affect the training and performance of the model. In this study, different seed experiments usually refer to using different random seeds to initialize the model's weight parameters under the same dataset and model settings, and using different standard deviation scaling methods to compare the performance differences between different seed experiments. The purpose of doing this is to test the stability and generalization ability of the model under different random initializations, and thus better evaluate the model's performance. By using different random seeds, the performance of the model can be more comprehensively evaluated, and the impact of incidental results caused by a particular random seed on experimental conclusions can be reduced.

In this study, three different seeds are used for experimentation. The detailed results are shown in Table 4. For seed 1, the weights are initialized using the "Normal" method, which initializes the weights using a standard normal distribution. This initialization method helps the model to converge faster and fit the data better. During initialization, the mean is kept at 0, and standard deviation scaling of 0.01, 0.02, and 0.03 are used, respectively.

For seed 2, the weights are initialized using the "Xavier" method, which initializes the weights using a normal distribution with mean 0 and variance $\sigma^2 = \frac{2}{n_{in}+n_{out}}$, where $n_{in}$ and $n_{out}$ are the input and output channel numbers of the weight matrix. The same standard deviation scaling of 0.01, 0.02, and 0.03 are used, respectively.

For seed 3, the weights are initialized using the "Kaiming" method, which is an orthogonal initialization method that initializes the weights based on an orthogonal matrix to help the model converge faster. The same standard deviation scaling of 0.01, 0.02, and 0.03 are used, respectively.

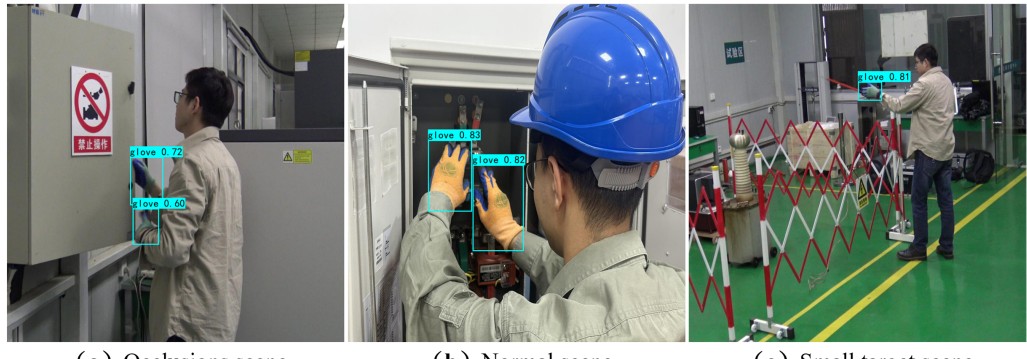

(**a**)Occlusions scene      (**b**)Normal scene      (**c**)Small target scene

**Figure 5**   **The detection effect of the CAPN proposed in different actual power operation sites.**

In summary, using different random seeds to verify the stability of the network is an effective experimental method that can help us more comprehensively evaluate the performance and stability of the model. In this article, it has been demonstrated by experiments that using the Normal weight initialization method with a standard deviation scaling of 0.01 can achieve the best results, with an average precision of 96.59%.

## Detection results

The results of the algorithm are visualized by testing glove detection in a set of real-world power operation scenarios. As shown in Fig. 5, the test data samples are collected from real power operation scenarios and are different from the original dataset. The detection results demonstrate that the proposed algorithm has good robustness and can accurately identify whether the operator is wearing gloves correctly in real-world power operation scenarios. For example, in (a), the transfer learning method employed in this article transfers hand features to the recognition of similar target gloves, allowing the network to better extract the features of hands wearing gloves. Thus, even with a small sample size of the dataset, gloves can still be accurately identified under occlusion. In (b), gloves can be accurately identified under good light and normal power operation conditions, as the features are more distinct. In (c), due to the attention mechanism introduced in the network structure, which can extract channel and spatial feature information at a deeper level, gloves can be accurately identified even under small target conditions, such as insufficient indoor ambient light and long distance from the target. Experimental results show that this method can improve recall rate and effectively enhance the recognition of small targets. Overall, the proposed algorithm can detect gloves in real time in various complex scenarios, effectively ensuring the personal safety of workers in power operation scenes.

## CONCLUSION

To reduce the risk of electrical safety accidents in power operation sites, a real-time glove detection network with transfer learning and an attention mechanism is proposed in this article. Two datasets are created: one for human hands and the other for gloves in electrical work sites. Transfer learning is employed to transfer the features of the hand model to

glove detection, in order to minimize missed detections in power operation sites. After experimental verification, the model's recall has greatly improved, effectively avoiding missed inspections of gloves. Additionally, an attention module called Channel-wise Attention and Position-aware Layer (CAP-Layer) is proposed, which combines channel space attention to effectively improve the detection precision of transfer learning and deepen the network's feature extraction. Experimental results on a small-scale dataset of 1,000 gloves demonstrate that the proposed network, referred to as CAPN, has effectively improved the AP of glove detection, with an overall average precision (AP) of 96.59%, compared with six classic target detection networks.

Although the proposed method achieved the best results, in some cases, such as when the operator clenches their fist wearing gloves, the detection performance may be suboptimal, leading to missed detections. Moreover, the current method can only recognize whether the operator is wearing gloves and cannot identify other potential safety hazards in the power scenario, such as whether the operator is wearing a safety belt or helmet. Therefore, in future work, we plan to expand the dataset to improve the recognition of various hand poses in the source domain and apply it to glove recognition. We also plan to extend the recognition categories, such as safety helmets and belts, and verify the network's robustness to these categories. In summary, the proposed real-time glove detection network using transfer learning and attention mechanisms has higher average precision in glove recognition compared to classical object detection networks. However, future work is still needed to improve the detection of gloves in special hand poses and other categories of related risks, further enhancing the robustness and average precision of the network.

### Funding

This work was supported by the National Natural Science Foundation of China (No. 62202346), Hubei Key Research and Development Program (No. 2021BAA042), the Open project of engineering research center of Hubei Province for clothing information (No. 2022HBCI01), the Wuhan applied basic frontier research project (No. 2022013988065212), MIIT's AI Industry Innovation Task unveils flagship projects (key technologies, equipment, and systems for flexible customized and intelligent manufacturing in the clothing industry), and the Hubei Science and Technology Project of safe production special fund (scene control platform based on proprioception information computing of artificial intelligence) (No. 62202346), the Hubei key research and development program (No. 2021BAA042), the Open project of engineering research center of Hubei province for clothing information (No. 2022HBCI01), and the Hubei science and technology project of safe production special fund (scene control platform based on proprioception information computing of artificial intelligence). The funders had no role in study design, data collection and analysis, decision to publish, or preparation of the manuscript.

### Grant Disclosures

The following grant information was disclosed by the authors:

National natural science foundation of China: 62202346.
Hubei key research and development program: 2021BAA042.
Open project of engineering research center of Hubei province for clothing information: 2022HBCI01.
Wuhan applied basic frontier research project: 2022013988065212.
MIIT's AI Industry Innovation Task unveils flagship projects.
Hubei science and technology project of safe production special fund (Scene control platform based on proprioception information computing of artificial intelligence): 62202346.
Hubei key research and development program: 2021BAA042.
Open project of engineering research center of Hubei province for clothing information: 2022HBCI01.
Hubei science and technology project of safe production special fund (scene control platform based on proprioception information computing of artificial intelligence).

## Competing Interests

The authors declare there are no competing interests.

## Author Contributions

- Feng Yu conceived and designed the experiments, analyzed the data, authored or reviewed drafts of the article, and approved the final draft.
- Jialong Zhu conceived and designed the experiments, performed the experiments, analyzed the data, performed the computation work, prepared figures and/or tables, authored or reviewed drafts of the article, and approved the final draft.
- Yukun Chen analyzed the data, performed the computation work, prepared figures and/or tables, authored or reviewed drafts of the article, and approved the final draft.
- Shuqing Liu performed the computation work, prepared figures and/or tables, and approved the final draft.
- Minghua Jiang performed the experiments, authored or reviewed drafts of the article, provide financial help and support for the project, and approved the final draft.

## Data Availability

The data is available at figshare:

- Zhu, Jialong (2023). glove dataset.zip. figshare. Dataset. https://doi.org/10.6084/m9.figshare.22138439.v1

- Zhu, Jialong (2023). hand-dataset. figshare. Dataset. https://doi.org/10.6084/m9.figshare.22138781.v1

The code is available at GitHub and Zenodo:

-https://github.com/wtujialongzhu/GloveDetNet

- wtujialongzhu, & SunYiGui. (2023). wtujialongzhu/GloveDetNet: GloveDetNet-main (computer). Zenodo. https://doi.org/10.5281/zenodo.8055708

## Supplemental Information

Supplemental information for this article can be found online at http://dx.doi.org/10.7717/peerj-cs.1558#supplemental-information.

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
