# Peer review of "CAPN: a Combine Attention Partial Network for glove detection"

_PeerJ Computer Science, doi:10.7717/peerj-cs.1558_

## Round 0.1 · original submission · Major Revisions

The paper proposed a target detection network called CAPN for real-time detection of protective gloves in the electric power scenario. Note that the English expression of the paper should be fully proofread, particularly in the experimental section.

Reviewer 1 ·

Basic reporting

In this paper, the authors proposed a target detection network called CAPN for real-time detection of protective gloves in the electric power scenario. The network uses a convolutional neural network-based method to design an attention mechanism that enhances the detection of small targets and improves the overall efficiency of the network using transfer learning. The various modules of the network are well presented in the figures. Overall, this paper is interesting and has practical application value, but there are some issues that need to be addressed:
(1)The English expression in the article need to be improved. In some places, the expression can be further revised. For example, in line 13 of the abstract, "low efficiency of manual supervision" can be replaced with "inefficiency of manual supervision." In line 14, "lack of dataset of gloves" can be replaced with "limited availability of glove datasets" to convey the meaning of the article better.
(2)In the related work section, more methods currently available for glove detection should be introduced.

Experimental design

The study effectively protects electric power personnel from dangerous situations using artificial intelligence. The research background has a certain significance, and the experimental results have verified the effectiveness of the proposed method. However, the following issues need to be addressed:
(1)Does this kind of study have never been attempted before? Justify this statement and give an appropriate explanation to do so in this paper.
(2)In the comparative experiments and ablation experiments, the innovative part of the transfer learning training method is not emphasized in the paper. Therefore, the specific training method of transfer learning should be emphasized in the experimental section.

Validity of the findings

The experimental design in this paper includes comparative experiments and ablation experiments, which effectively demonstrate the validity of the proposed method. The method theoretically achieves robustness for various gloves. However, the following issues need to be addressed:
(1)In line 302, it is necessary to clarify that adding transfer learning in the ablation experiment solved the problem of a small data sample size.
(2)The conclusion section should summarize the findings related to the original problem and describe the methods and techniques used to ensure the validity of the results.

Additional comments

None

Cite this review as

Reviewer 2 ·

Basic reporting

I have carefully read your paper and would like to offer the following comments and suggestions regarding its quality and acceptability:
(1)Your paper has a reasonable structure, but it needs further improvement in language expression and style. Some sentences are not clear enough and require more details and background information to help readers understand your research.
(2)The references cited in your paper also meet the requirements, citing the latest and most relevant literature as much as possible to ensure that your references and article content remain in sync and reflect your latest understanding of research in this field.

Experimental design

The experimental section demonstrates the effectiveness of the proposed method and improves the accuracy of glove detection.

Validity of the findings

I make no judgement on the correctness of the work, only on its suitability according to general criteria, especially containing significant new experimental results and being recognized as a very important contribution to the literature.

Cite this review as

Reviewer 3 ·

Basic reporting

The paper proposes a glove detection algorithm using video surveillance to detect whether operators are wearing gloves correctly in real time, addressing the issue of accidents in electric power operation sites due to operators not wearing safety protection equipment. The paper uses transfer learning and an attention mechanism to enhance detection accuracy. It proposes the Combine Attention Partial Network (CAPN) network based on convolutional neural networks to improve the extraction of deep feature information and recognition accuracy.
The paper is well-written, and the proposed algorithm is exciting and innovative. The paper explains the key ideas behind the algorithm and the experimental results demonstrating its high performance in terms of detection accuracy and computational speed. However, the paper could benefit from additional information on the dataset used and the evaluation metrics used to measure the algorithm's performance.
Moreover, it would be helpful if the paper included a discussion of the potential limitations of the proposed algorithm and the future work required to overcome these limitations. This would enhance the paper's contribution to the field and provide insight into the areas that require further research.
Overall, the paper is a valuable contribution to the field and presents an innovative solution to a significant problem. The proposed algorithm can potentially improve safety in electric power operation sites by detecting whether operators are wearing gloves correctly in real-time. The paper could further enhance its contribution to the field with some additional information and discussion.

Experimental design

The experimental design is well constructed. However, some must be empowered. For example, it could be better if the authors reported experiments with different seeds and the accuracy mean.

Validity of the findings

The work proposes valid findings. I have to be honest with the authors. The title must be changed because it seems that they talk about GLove in NLP.

Cite this review as

Reviewer 4 ·

Basic reporting

I reviewed your work titled "GloveDetNet: glove detection network with transfer learning and attention mechanism" in detail.

Experimental design

The points that I see missing in the study are presented below in the articles. It is also important that you mention the results of the proposed model in the Abstract section. Studies on the subject in the literature can be expanded. I recommend you review the related work "https://ieeexplore.ieee.org/document/9581987". Why didn't you use sample images from the dataset? The detection results in the section is very short. The relevant section needs to be detailed. In the study, the words in the we style were used in large numbers. The use of such words is not welcome in the study. If the study is supported with confusion matrices, it will look better. I don't think the Data Augmentation header is correct. It is not correct to detail, for example, pre-trained models under this title. The proposed model should be presented concisely and more regularly. In particular, why this model was presented and its contributions to the literature should be clearly summarized. A paragraph about the proposed model should also be added at the end of the Introduction section.

Validity of the findings

Limitations of the study should be addressed.

Additional comments

It is important to review the spelling errors in the study.

Cite this review as

---

## Round 0.2 · Minor Revisions

I suggest the authors should further improve the paper in accordance with the remaining comments.

Reviewer 1 ·

Basic reporting

no comment

Experimental design

no comment

Validity of the findings

no comment

Additional comments

All my concerns have been addressed. I recommend this paper for publication.

Cite this review as

Reviewer 3 ·

Basic reporting

The following manuscript has clear improvements. However, some things still need to be improved. First, I think the authors could consider an Epistemological Study of Knowledge in NN-based models. This could turn the work in a positive direction. Secondly, there are still inaccuracies in the exps. The methods are not very understandable, and I had to read a lot to understand them.

Experimental design

It sounds good.

Validity of the findings

I confirm the above. there is quality and there are results.

Cite this review as

---

## Round 0.3 · accepted · Accept

I confirm that the authors have addressed all of the reviewers' comments.

Reviewer 4 ·

Basic reporting

.

Experimental design

.

Validity of the findings

.

Additional comments

.

Cite this review as